# Susceptibility of Ocular Surface Bacteria to Various Antibiotic Agents in a Romanian Ophthalmology Clinic

**DOI:** 10.3390/diagnostics13223409

**Published:** 2023-11-09

**Authors:** Aurelian Mihai Ghita, Daniela Adriana Iliescu, Ana Cristina Ghita, Larisa Adriana Ilie

**Affiliations:** 1Department of Physiology, “Carol Davila” University of Medicine and Pharmacy, 8 Eroii Sanitari Bld., 050474 Bucharest, Romania; ghita.amg@gmail.com; 2Ophthalmology Department, Bucharest University Emergency Hospital, 169 Splaiul Independenței Street, 050098 Bucharest, Romania; 3Ocularcare Ophthalmology Clinic, 128 Ion Mihalache Bld., 012244 Bucharest, Romania; eyeana.ghita@gmail.com (A.C.G.); larisaadriana1907@gmail.com (L.A.I.)

**Keywords:** ocular infections, bacteria contamination, antibiotic resistance

## Abstract

Periodic assessment of bacterial contamination is necessary as it allows proper guidance in cases of eye infections through the use of appropriate antibiotics. Due to the extensive use of antibiotic treatment, many strains of the microbiota that cause infections are resistant to the usual ophthalmic antibiotics. The present study provides an updated assessment of the susceptibility of Gram-positive and Gram-negative bacteria found on the ocular surface to the most commonly used antibiotic agents in patients undergoing cataract surgery. A total of 993 patients were included in the study with ages between 44 and 98 years old. Conjunctival cultures were collected 7 days before cataract surgery. The response of Gram-positive and Gram-negative bacteria to various antibiotic classes, such as glycopeptides, cephalosporins, carbapenems, fluoroquinolones, aminoglycosides, phenicols, tetracyclines, rifamycins, macrolides and penicillins, was assessed. From the tested antibiotics, vancomycin had 97.8% efficacy on Gram-positive bacteria. In the cephalosporin category, we observed a high level of resistance of the cefuroxime for both Gram-positive and negative bacteria. Antibiotics that have more than 90% efficacy on Gram-positive bacteria are meropenem, imipenem, netilmicin, amikacin and rifampicin. On Gram-negative bacteria, we found 100% efficacy of all tested fluoroquinolones, i.e., aminoglycosides (except for tobramycin), doxycycline, azithromycin, clarithromycin and chloramphenicol. The current study illustrates patterns of increased resistance in certain bacteria present on the ocular surface to some of the commonly used antibiotics in ophthalmological clinical practice. One such revealing example is cefuroxime, which has been highly used as an intracameral antibiotic for the prevention of bacterial endophthalmitis after cataract surgery.

## 1. Introduction

Periodic assessment of bacterial contamination is necessary as it allows proper guidance in the use of antibiotics in the case of eye infections. Due to the extensive use of antibiotic treatment, many strains of the microbiota are resistant to usual ophthalmic antibiotics. Other risk factors, such as suboptimal dosing and patient noncompliance, often culminate in subtherapeutic antibiotic concentrations, fostering an environment conducive to resistance selection. Moreover, horizontal gene transfer mechanisms, such as conjugation, transformation, and transduction, have further accelerated the dissemination of resistance determinants amongst bacterial populations [1,2].

The present study provides an updated assessment of the susceptibility to commonly used antibiotic agents of Gram-positive and Gram-negative bacteria found in patients that had ocular surface contamination before undergoing cataract surgery.

Although rare, eye infections are vision-threatening conditions that have a highly negative potential on the quality of life of the patient. Endogenous or exogenous endophthalmitis remains one of the most redundant eye infections. Postoperative endophthalmitis is a major challenge that can result in eye loss despite proper treatment. Even though the most frequent cause of iatrogenic endophthalmitis is cataract surgery, its percentage from the total number of cases has dropped, as the number of intravitreal injection procedures has increased significantly [3]. Of the microbial causes of endophthalmitis, 85.1% are represented by Gram-positive agents, 10.3% by Gram-negative bacteria and the rest of the percentage by other types of microorganisms [4]. The most frequent pathogens found on the ocular surface are *Staphylococcus aureus*, *coagulase-negative Staphylococcus* and *Corynebacterium* [5,6]. The usual bacteria in the conjunctival microbiota and on the ocular surface are the most common etiological agents found in iatrogenic eye infections. 

The method of prescription of antibiotic medication varies depending on the geographical region, but medical practice usually implies prescribing a broad-spectrum antibiotic. The most-often used topical antibiotics are tobramycin, ciprofloxacin, ofloxacin, moxifloxacin, chloramphenicol, gentamicin, azithromycin, and erythromycin [5,7,8]. Tobramycin is more frequently used in children, while fluoroquinolones are usually prescribed in adults [9]. Fluoroquinolones, the representative drug class for treating ocular infections, have shown diminishing efficacies, especially against *methicillin-resistant Staphylococcus aureus* (MRSA) and *Pseudomonas aeruginosa* [10,11,12]. The mechanism often implies mutations in the DNA gyrase and topoisomerase IV genes, rendering them unreceptive to the antibiotic’s action [13,14]. Aminoglycosides, another therapeutic mainstay, are being inhibited by resistance mechanisms, most notably the enzymatic inactivation pathways [15].

By knowing as accurately as possible the bacterial spectrum and antibiotic sensitivity, a therapeutic strategy can be established to prevent bacterial contamination by using pre-, peri- and postoperative antibiotics. In cases of eye infection, broad spectrum antibiotics may be used until bacterial cultures and antibiograms are obtained.

## 2. Materials and Methods

A total of 993 patients were included in the study, with ages between 44 and 98 years old. The prospective study was conducted among patients attending Ocularcare Ophthalmology Clinic, Bucharest, Romania who were undergoing cataract surgery on their first eye. Ocular pathogens were collected between January 2022 and August 2023. Conjunctival swab was collected 7 days before surgery from the eye undergoing surgical intervention for cataract. In case of a positive result, an antibiogram was performed. The bacterial cultures and antibiograms of the patients were performed in laboratories approved by the Romanian Accreditation Association (RENAR) and that fulfilled the requirements SR EN ISO 15189. All laboratories used the same bacterial identification and antimicrobial sensibility testing technique. Only those antibiotics that were tested in all laboratories were considered for analysis. We obtained informed consent from all patients involved in the study. Personal data of the patients were kept in strict confidence, as well as any patient-related results. The conjunctival ocular samples that were taken are part of the standard medical practice that patients undergo before cataract surgery. The results were displayed as a percentage for each type of bacterial response to the antibiotic. The number of samples tested for the antibiotic was also considered out of the total number of positive samples. Sample collection was performed using a sterile cotton-tipped swab; a trained nurse gently swabbed 2-fold the conjunctival sac, taking care not to touch the eyelashes, eyelids, or any other surfaces. The swab was placed into a sterile Stuart’s transport medium to maintain bacterial viability. All collected samples were tested for microorganism species and analyzed through the broth microdilution technique. Bacterial isolated were inoculated onto blood agar and agar chocolate plates and were incubated in a 5% CO_2_ atmosphere for 24 h at 35–37 degrees Celsius to promote bacterial growth. After 24 h of incubation, the plates were examined for bacterial growth. Preliminary identifications were made based on colony characteristics such as size, color and morphology. Following the first step, Gram staining, subculture, biochemical analysis and API (Analytical Profile Index) strips were performed for the identification of the bacterial species. Bacteria were identified according to the conventional methods used for each microorganism class in accordance with Clinical and Laboratory Standards Institute (CLSI) protocols [16].

The antimicrobial susceptibility testing was performed using broth microdilution technique, which uses multi-well plates containing increasing concentrations of antibiotics in a broth medium. The broth microdilution technique determines the Minimum Inhibitory Concentration (MIC) of an antimicrobial agent against a particular microorganism. The MIC is the lowest concentration of the antimicrobial that inhibits visible growth of the microorganism after a specified incubation time. For the preparation of antimicrobial dilutions, decreasing concentrations of the antimicrobial agent were prepared in Mueller-Hinton growth medium. This was performed in 96-well microtiter plates, each well containing a different concentration of the antibiotic. The bacterial suspension, matching a 0.5 McFarland standard for turbidity, was diluted to achieve the desired concentration, ensuring that each well of the microtiter plate would contain a standardized number of bacteria. The microtiter plate was incubated at 35–37 °C for 24 h. After incubation, each well was examined for visible bacterial growth. The bacteria susceptibility to antibiotics was interpreted as follows: resistant, intermediate, and sensitive by following the response compared to MIC. A sensitive result implies that the organism is inhibited by the serum concentration of the drug that is achieved using the usual dosage; an intermediate result implies that the organisms are inhibited only by the maximum recommended dosage for each antibiotic; and a resistant result implies that the organisms are resistant to the usually achievable serum drug levels. When an intermediate sensitivity response of the bacteria to a certain antibiotic is present, the therapeutic effect of the medication is unreliable. These interpretative standards have been established by the CLSI procedures. For each bacteria category, the response to several antibiotics was evaluated. The susceptibility of bacteria was tested for various antibiotics from the following classes: glycopeptides (vancomycin), cephalosporins (ceftriaxone, cefuroxime, cefazolin), carbapenems (meropenem, imipenem), fluoroquinolones (moxifloxacin, levofloxacin, ofloxacin, ciprofloxacin), aminoglycosides (tobramycin, netilmicin, amikacin, kanamycin, gentamicin), phenicol’s (chloramphenicol), tetracyclines (tetracycline, doxycycline), rifamycin’s (rifampicin), macrolides (azithromycin, clarithromycin, erythromycin) and penicillin’s (ampicillin, amoxicillin). Data were stored in an Excel table and statistically analyzed in IBM SPSS Statistics.

## 3. Results

From all positive tested conjunctival samples, 86.33% were identified as Gram-positive bacteria, while 13.66% were Gram-negative. For the Gram-positive bacteria, *methicillin-sensitive Staphylococcus aureus* had the highest prevalence (55.9%), followed by *coagulase-negative Staphylococcus* (21.73%), *methicilin-resistent Staphylococcus aureus* (3.1%) and *Streptococcus* (1.86%) (Figure 1). Other identified Gram-positive bacteria were *Enterococcus* (2.48%) and *Corynebacterium macginleyi* (1.24%). From the Gram-negative bacteria, the highest prevalence was found for *Klebsiella* (4.34%), *Proteus* (3.72%) and *Pseudomonas aeruginosa* (1.86%). Other less frequent Gram-negative bacteria were *Haemophilus* spp. (1.24%), *Escherichia coli* (1.24%), *Serratia marcescens* (0.62%) and *Enterobacter* spp. (0.62%). For each bacteria, the isolates resistance patterns are summarized in Table 1.

From the tested antibiotics, vancomycin had 97.8% efficacy on Gram-positive bacteria. It has not been tested on Gram-negative bacteria as it is not on their action spectrum. Of the Gram-positive microorganisms, 2.2% had an intermediate response for vancomycin. In the studied group, no resistance to vancomycin was detected. In the cephalosporin category, we observed a high level of resistance of the highly used cefuroxime for both Gram-positive and negative bacteria (Table 2). A total of 76.9% of the Gram-positive and 45.5% of the Gram-negative were sensitive to cefuroxime. In contrast, Gram-negative bacteria were 100% sensitive to ceftriaxone. Very few bacteria were tested for cefazolin, a first-generation cephalosporin antibiotic; nevertheless, they manifested very high resistance. Other effective antibiotic agents are from the carbapenem class. Over 90% of the Gram-positive and 100% of the Gram-negative bacteria were sensitive to meropenem and imipenem. Other antibiotics that have more than 90% efficacy on Gram-positive bacteria are netilmicin and rifampicin (Table 3). Moderate efficacy on Gram-positive microorganisms (80% to 90% of the bacteria were sensitive to antibiotics) was evident for fluoroquinolones (moxifloxacin, levofloxacin, ofloxacin, ciprofloxacin) and chloramphenicol. Decreased therapeutic effect on Gram-positive (less than 80% of the bacteria respond to antimicrobial treatment) was seen for cephalosporins, penicillins, aminoglycosides (except for netilmicin), macrolides and tetracyclines.

On Gram-negative bacteria, we found 100% efficacy of all tested fluoroquinolones (moxifloxacin, levofloxacin, ofloxacin, ciprofloxacin), carbapenems (meropenem, imipenem), aminoglycosides (amikacin, kanamycin, gentamicin), chloramphenicol, some of the macrolides (azithromycin, clarithromycin) and doxycycline (Table 4). The exception regarding the efficacy of aminoglycosides on Gram-negative was observed for tobramycin, which was sensitive for 90% of the bacteria. Decreased efficacy on Gram-negative bacteria was observed for antibiotics like tetracycline (85.7%) and rifampicin (66.7%). Far less efficient on Gram-negative bacteria are antimicrobials from the penicillin’s class.

## 4. Discussion

The susceptibility of bacteria to antibiotics in our study was similar to that reported in the literature but not for all types of antibiotic agents. Gram-positive bacteria responded well to vancomycin in all tested cases, except one were the bacteria had intermediate sensitivity to the antimicrobial. This fact makes it extremely valuable in the treatment of bacterial endophthalmitis administrated through intravitreal injections. Nevertheless, intracameral administration of vancomycin as a prevention strategy for endophthalmitis has been associated with the development of retinal vasculitis [17,18,19]. This prevention treatment is mainly practiced in Australia and less common in European countries [19]. Vancomycin does not have an antimicrobial effect on Gram-negative bacteria and is usually associated with other antibiotic classes like aminoglycosides or ceftazidime in the treatment of bacterial endophthalmitis [20]. Adjunctive systemic antibiotics that achieve intravitreal therapeutic levels can also be administered, the best-known agents being meropenem, moxifloxacin and linezolid [21]. Very rare Gram-positive bacteria, such as *coagulase-negative Staphylococcus* and *Enterococcus*, have been found to express reduced susceptibility to vancomycin in bacterial endophthalmitis [22,23,24]. For such rare cases of vancomycin resistant *Enterococci*, intravitreal administration of linezolid was attempted but there are still concerns regarding the safety and adverse reactions [25]. Nevertheless, the association of vancomycin 1.0 mg/0.1 mL together with ceftazidime 2.25 mg/0.1 mL remains the most frequently used therapeutic scheme in the management of bacterial endophthalmitis [26]. This combination covers a broad-spectrum of microbial agents. In allergic patients to ceftazidime, vancomycin can be associated instead with amikacin 0.4 mg/0.1 mL [27,28,29]. Some reports suggested an even greater susceptibility of Gram-negative bacteria to amikacin than ceftazidime [30]. In the present study, we found no resistance of Gram-negative bacteria to amikacin. Even though earlier endophthalmitis vitrectomy studies showed that intravitreal antibiotic injections should be the first line of treatment for cases of endophthalmitis following cataract surgery, newer studies show promising result and better visual outcomes as well as fewer total procedures when primary vitrectomy is performed [27,31,32].

Evaluating the response to cephalosporins, cefuroxime (which is often used intracameral as prevention therapy against bacterial infections after cataract surgery) displayed a moderate efficiency on Gram-positive bacteria and a weak therapeutic effect upon Gram-negative bacteria. Only 76.9% of the Gram-positive and 45.5% of the Gram-negative bacteria were sensitive to cefuroxime. Prophylactic intracameral cefuroxime has been used for over 20 years in the prevention of endophthalmitis after cataract surgery [33]. When intraoperative anterior chamber irrigation of cefuroxime was compared to balanced salt solution, the rate of postoperative endophthalmitis after cataract surgery decreased seven-fold [34]. Multiple studies also showed a decreased incidence of endophthalmitis after introducing cefuroxime in their surgery protocol [35,36,37,38,39]. Even though reports of routinely used intracameral cefuroxime for the prevention of postoperative endophthalmitis show good results, this protocol must be regularly reevaluated, and signs of antimicrobial resistance must be thoroughly reported [20,40]. Despite the fact that postoperative endophthalmitis incidence after usage of intracameral cefuroxime was reported to be only 0.033%, newer protocols that tested intracameral moxifloxacin or vancomycin showed even lower incidences (0.015% and 0.01%). This difference could be an indication of increased bacterial resistance to cefuroxime [41]. A bacteriology profile and sensitivity to cefuroxime study conducted on a 20-year period showed an important shift to resistant organism. From the evaluated 20 years’ timeline, in the first period of the study, endophthalmitis after cataract surgery was mainly caused by *coagulase-negative Staphylococci*, *Staphylococcus aureus* and *Streptococci*, while in the second period of the study, after the introduction of intracameral cefuroxime, the incidence of *Enterococci* and cefuroxime-resistant bacteria increased significantly [38]. Other studies also reported endophthalmitis with cefuroxime-resistant bacteria after phacoemulsification [42,43,44,45]. Some clinical trials have suggested that topical antibiotics used in addition to intracameral cefuroxime lowers the chance of post-operative endophthalmitis when compared to cefuroxime injection alone [46]. Other studies with low or moderate risk of bias reported no difference between the two protocols [41]. Surgeons must take into consideration that intraoperative administration of cefuroxime can reduce the bacterial infection rate but also should not ignore the possibility of Gram-positive resistance development or the weak effect of the antibiotic on Gram-negative bacteria [47]. The pattern of increasing resistance is concerning for the population of Romania, given the critical role of cefuroxime in preventing postoperative endophthalmitis. Continuous monitoring of resistance patterns is essential. Regular updates can ensure timely shifts in prophylactic strategies when needed. The emerging resistance of bacterial strains to cefuroxime underscores the dynamic nature of microbial evolution and the need for ongoing vigilance in clinical practice. Embracing robust surveillance systems, antibiotic stewardship, and research into alternative prophylactics will be vital in ensuring optimal outcomes for patients undergoing intraocular surgeries. Ceftriaxone, another antibiotic evaluated from the cephalosporin class, showed a moderate effect on Gram-positive bacteria (only 72.72% of the bacteria were sensitive to the antibiotic) but an excellent effect on Gram-negative bacteria (100% of the bacteria were sensitive to the antibiotic). In the treatment of endophthalmitis, ceftriaxone (at concentrations more than 50 mg/dose) cannot be administered as an intravitreal medication due to ocular toxicity but can by utilized as a systemic adjuvant therapy [48,49]. Systemic delivery of ceftriaxone will produce intravitreal antibiotic levels, which inhibit *Streptococci* and *Enterobacteriaceae* but not *Staphylococcus aureus* [21,50].

Our study showed that carbapenems are a highly efficient antibiotic class against both Gram-positive and Gram-negative bacteria (100% of the Gram-negative samples and over 90% of the Gram-positive samples were sensitive to meropenem and imipenem). For ocular infections, carbapenems are mainly used as an adjuvant systemic therapy in the treatment of endophthalmitis as they achieve efficient vitreous concentrations that are well above the necessary breakpoints of the Gram-positive and negative bacteria [51]. On the other hand, despite reports of proper intravitreal concentration of meropenem after systemic delivery, some studies show no additional benefit in visual outcome when compared to conventional systemic antibiotics in the treatment of postoperative endophthalmitis [52]. Experimental models that evaluated topical meropenem in comparison to other antibiotic treatments showed promising result for the treatment of ocular infections with *Pseudomonas* [53]. Studies also revealed that administration of topical meropenem as bacterial keratitis treatment has good corneal penetration as well as low toxicity. In addition to *Pseudomonas*, the efficacy of meropenem was proved on *Staphylococcus aureus*, *coagulase-negative Staphylococci*, *Streptococcus* and *Enterobacteriaceae* infections [54]. In vitro studies have shown higher susceptibility of both Gram-positive (including *methicilin-resistent coagulase-negative Staphylococcus*) and Gram-negative bacteria to imipenem when compared to linezolid, tigecycline or fluoroquinolones [55]. Also, in experimental models, attempts to treat post-traumatic *Pseudomonas* endophthalmitis through intravitreal injections of meropenem have shown promising results when compared to intravitreal ceftazidime [56]. An important aspect to be noted is that meropenem, unlike ceftazidime, requires three intravitreal doses. Even though, meropenem is a potent antibiotic, very few studies analyze its therapeutic effect, adverse reactions on the retina and other ocular structures and the remanence time in the vitreous. The latter point appears to be one of the major problems that must be further studied, as meropenem has been reported in one study to have a half-life (t(1/2)) in the vitreous of only 2.6 h [57]. To review, topical carbapenems emerge as a potent treatment for severe ocular surface infections, as its efficacy has been demonstrated to be great on both Gram-positive and -negative organisms, but few studies are available that assess any adverse effects, while treatment through intravitreal administration faces the problem of high vitreous washout dynamics.

In the present study, the therapeutic response to fluoroquinolones has been found to be moderate for Gram-positive bacteria (80.6%, 85.2%, 82.1% and 82.8% sensitive to moxifloxacin, levofloxacin, ofloxacin and ciprofloxacin) and excellent for Gram-negative organisms (100% sensitive for all four tested agents). Adequate efficiency on microorganisms of fluoroquinolones makes them appropriate as prevention treatment before surgery and decreases bacterial contamination. Nevertheless, prolonged use of antibiotics like fluoroquinolones induces antimicrobial resistance, which has become increasingly prevalent in situations of ocular infections [58,59,60,61]. The use of fluoroquinolones empirically as a broad-spectrum antibiotic for ocular infections as well as excessive administration in the perioperative period for procedures like intravitreal injections have increased their antimicrobial resistance. This is supported by results that reveal an increased risk of endophthalmitis after fluoroquinolone antibiotic prophylaxis due to the selection of resistant conjunctival microbiota [62,63]. Resistance patterns depend not only on the type of antibiotic but also the regional prescription habits, with great variability between countries [64]. In our study, increased resistance was observed for fluoroquinolones among Gram-positive bacteria, moxifloxacin being the least effective antimicrobial, even though it is a fourth-generation fluoroquinolone. This trend was highlighted also by other recent studies [60,65]. Moreover, studies show that a one-month prophylactic treatment of levofloxacin after cataract intervention yields fluroquinolone-resistant microorganisms [66]. While reports on ciprofloxacin-resistant Gram-positive bacteria have been also observed, other fluoroquinolones like besifloxacin have shown good therapeutic result in ocular infections with *Staphylococcus aureus* or *methicillin-resistant Staphylococcus epidermidis* [67]. Other studies likewise observed no difference in the final outcomes between moxifloxacin and fortified vancomycin in the treatment of *methicillin-resistant Staphylococcus aureus* [68]. For Gram-negative bacteria, fluroquinolones are extremely useful in the prevention of endophthalmitis before surgery. Fluoroquinolones were considered, in general, not suitable for intravitreal injections due to their toxicity. Still, few cases have been published showing good results after intravitreal injections of moxifloxacin for the treatment of acute post-operative endophthalmitis, or for the prevention of it, following cataract surgery [69,70]. While some scientific papers reveal the tendency of moxifloxacin to exhibit increased antimicrobial resistance, others provide evidence that intracameral moxifloxacin could be a proper alternative for the prevention of endophthalmitis after surgery. As highlighted previously, some of the regularly used intracameral antibiotics (especially cefuroxime) raise concerns about bacterial resistance. Therefore, moxifloxacin is being investigated as a more adequate option. When compared to topical antibiotics given after surgery, intracameral moxifloxacin has shown a reduction of the endophthalmitis incidence rate by up to four-fold. Nevertheless, when compared to intracameral cefuroxime, statistics indicated no significant benefit [70,71,72,73,74]. Comparing our result to other regional reports, the overall resistance rate of fluoroquinolones (for both second and fourth generation) was greater in countries like India, the United States or Egypt [65,75,76]. Nevertheless, for the Indian population, the highest bacterial resistance was reported for ciprofloxacin, while moxifloxacin even showed a reduction of the resistance rates when the 2009–2012 period was compared to 2017–2020 [65]. Also, when comparing different drugs among the same class, in the US, moxifloxacin had the lowest resistance rate [76]. On the other hand, in countries like Egypt where the prevalence of methicilin-resistant infections is high, the rates of moxifloxacin resistance were also high [75].

Aminoglycosides have long been used for the treatment of ocular infections both as a topical and as intravitreal medication. In our study, Gram-positive bacteria showed quite high resistance to aminoglycosides, mostly for tobramycin and kanamycin (only 72.9% and 62.5% of the bacteria, respectively, were sensitive to them). Netilmicin was the most effective antibiotic tested on Gram-positive organisms (94.7%). Regarding the Gram-negative bacteria, tobramycin was the least effective antimicrobial, while 100% were sensitive to amikacin, kanamycin, and gentamicin. Overall, this antibiotic class showed moderate potency on Gram-positive microorganisms and a better efficacy on Gram-negative microorganisms. Retrospective studies have also confirmed a trend of antimicrobial resistance of aminoglycosides to Gram-positive bacteria, especially *Staphylococcus aureus* and *coagulase-negative Staphylococcus*, which are among the most frequent pathogens involved in ocular infections [77,78]. Not all studies reported a progressive increase in tobramycin bacterial resistance. This aspect is highly dependent on the prescription pattern of the region. In the US, the Antibiotic Resistance Monitoring in Ocular Microorganisms (ARMOR) surveillance study reported a small decrease in the *Staphylococcus aureus* resistance rates to tobramycin, at least for the timeline 2009–2016 [79]. While other antibiotic classes penetrate the aqueous humor after topical administration, netilmicin and tobramycin do not reach detectable concentrations. Thus, they are not adequate for intraocular infections when used as topical treatments [80]. For the prophylaxis of endophthalmitis in patients that underwent cataract surgery, topical aminoglycosides were ineffective when compared to gatifloxacin [81]. On the other hand, gentamicin attained good out-turn in reducing the incidence of endophthalmitis when injected into the surgical perfusion solution. Still, some results indicate that gentamicin may ease the development of resistant strains of *Enterococcus* [82]. For the prophylaxis of endophthalmitis after intravitreal injections with anti-VEGF, aminoglycosides are not indicated, especially when they are associated with corticosteroids, as the risk of infection is actually higher for the later compared to no antibiotic prophylaxis [63,83,84]. Aminoglycosides can be used as intravitreal treatment being both effective and with a much lower toxicity level compared to fluoroquinolones. Nevertheless, caution still must be exercised regarding the retinal toxicity. 

We observed a good efficacy of chloramphenicol for both Gram-positive and negative bacteria. We cannot assess with high accuracy the efficiency on Gram-negative bacteria due to the small sample of culture tests. Nevertheless, chloramphenicol remains a viable option as prevention treatment before and after ocular surgery and is still an active antibiotic against ocular infections [85,86]. It has also shown limited bacterial resistance when compared to other broadly used antibiotics like fluoroquinolones [87,88,89]. A 30-year study, that focused on antimicrobial resistance trends showed that, overall, chloramphenicol is one of the most effective antibiotics against bacterial ocular infections [60,90]. Even more severe eye infections induced by *methicillin-resistant Staphylococcus aureus* showed a generally good response to chloramphenicol and an uncommon bacterial resistance [88,91,92,93]. For chloramphenicol-resistant microorganisms, newer derivates like chloramphenicol-borate have emerged as a potential new antibiotic treatment [94]. In recent years, the use of chloramphenicol has been reestablished after a period of reduced usage due to reports of aplastic anemia adverse reactions [95,96]. 

Tetracyclines have been used most commonly in ophthalmological practice as topical or oral antibiotics for the treatment of *Chlamydia trachomatis*, which is a Gram-negative bacterium that determines one of the leading infectious causes of blindness in the world [97]. No significant difference was detected for the treatment outcome of active trachoma at 3 and 12 months when oral and topical treatments were compared. On the other hand, in the present study, we reported 14.3% resistance of Gram-negative microorganisms to tetracycline and zero to doxycycline. Increased resistance was observed for Gram-positive microorganisms (47.1% and 47.4% of the reported samples were sensitive to tetracycline, and doxycycline, respectively). This trend of increased bacterial resistance is in accordance with other recent studies [98,99,100,101]. Nevertheless, reports have shown that for certain bacterial species like *coagulase-negative Staphylococci* the resistance rates to tetracycline have dropped [78]. Resistance report of Chlamydia trachomatis to tetracycline as scarce, even though they seem to be increasing [97,102,103]. In addition to the antibacterial effect, tetracyclines exhibit anti-inflammatory properties that render them appropriate as an adjunctive therapy for posterior blepharitis or rosacea-associated ocular manifestations [104,105,106,107,108,109].

Macrolides (azithromycin, erythromycin and clarithromycin being the most prescribed) have been used in ophthalmological practice for the treatment of *Chlamydia trachomatis* and *Gonococcus* ocular infections. They have been administrated in the form of oral tablets, ophthalmic ointment, topical gel, or eye drop solutions [110,111,112,113]. In the present study, we found no resistance of the Gram-negative bacteria to azithromycin and clarithromycin. Even though macrolides have been characterized as having a broad antimicrobial spectrum, we found reduced sensitivity of Gram-positive bacteria to this antibiotic class, clarithromycin being the most effective, while azithromycin was the least efficient. Penicillins are not usually used for ocular infections due to the increased bacterial resistance. 

The limitations of the study are the inhomogeneous testing settings (even though all of the laboratories performed the antimicrobial susceptibility testing by using broth microdilution technique, not all of them used the same antibiotic set), and the relatively smaller number of Gram-negative bacteria tested. The lower number of Gram-negative samples is due to the fact that Gram-positive contamination of the ocular surface and lid margin is more frequent. 

The novelties and contributions of the present study are providing an updated picture of the dynamics of antibiotic treatments in bacterial contamination of the ocular surface and an increased awareness of the antimicrobial resistance that cefuroxime developed due to long-term usage as an intracameral prevention treatment for endophthalmitis. Also, we report an increased bacterial resistance to moxifloxacin, another frequently used broad-spectrum antibiotic. Knowledge of the regional rates of bacterial resistance to antibiotic is extremely important for proper management of ophthalmologic infections. It is crucial for clinicians to be aware of the latest local resistance data and adjust their prescribing patterns accordingly. Also, the current study illustrates the real-life setting and offers guidance for the prevention and treatment of ocular infections. In conclusion, the growing resistance patterns of bacteria in ocular infections signal a need for a paradigm shift in therapeutic strategies. Fostering prudent antibiotic use and investing in research for novel antimicrobial agents will be pivotal in navigating this challenge.

## Figures and Tables

**Figure 1 diagnostics-13-03409-f001:**
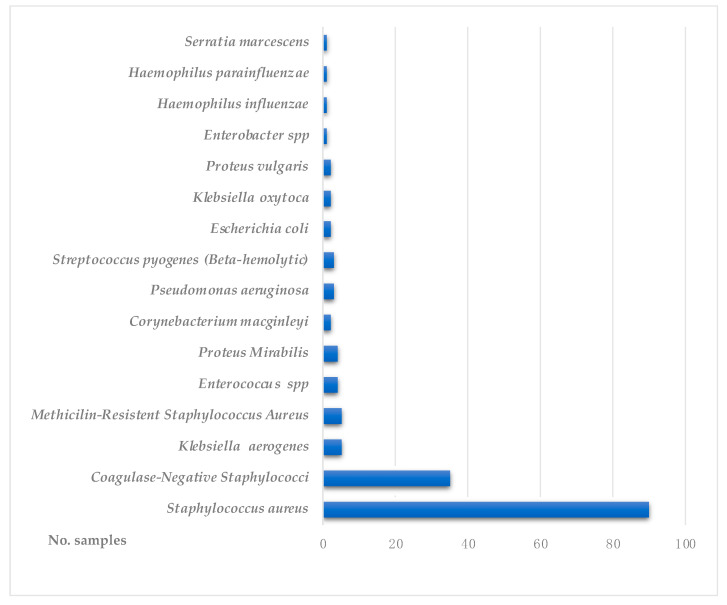
Distribution of bacterial isolates from the ocular surface of the tested subjects.

**Table 1 diagnostics-13-03409-t001:** Resistance patterns to antibiotics of different bacterial isolates.

Bacterial Isolates Species	No.	AS	Commonly Tested Antimicrobial, No. (%)											
			AMP	AMX	CLR	CXM	CIP	ERY	GEN	LVX	OFX	TET	TOB	VAN
*Staphylococcus aureus*	90	S	3 (15.7)	6 (46.15)	12 (52.17)	19 (82.6)	54 (84.37)	34 (47.22)	55 (82.08)	29 (82.85)	15 (75)	24 (52.17)	25 (75.75)	26 (100)
		I	0 (0)	0 (0)	0 (0)	0 (0)	5 (7.81)	4 (5.55)	2 (2.98)	4 (11.42)	1 (5)	2 (4.34)	4 (12.12)	0 (0)
		R	16 (84.2)	7 (53.84)	11 (47.8)	4 (17.39)	5 (7.81)	34 (47.220	10 (14.92)	2 (5.71)	4 (20)	20 (43.47)	4 (12.12)	0 (0)
*Coagulase-Negative Staphylococci*	35	S	1 (12.5)	3 (60)	5 (45.45)	10 (76.92)	19 (79.16)	7 (30.43)	21 (70)	12 (85.71)	5 (100)	7 (41.17)	9 (69.23)	11 (91.67)
		I	0 (0)	0 (0)	0 (0)	1 (7.69)	2 (8.33)	2 (8.69)	1 (3.33)	0 (0)	0 (0)	1 (5.88)	3 (23.07)	1 (8.33)
		R	7 (87.5)	2 (40)	6 (54.54)	2 (15.38)	3 (12.5)	14 (60.86)	8 (26.67)	2 (14.2)	0 (0)	9 (52.94)	1 (7.69)	0 (0)
*Klebsiella* spp.	7	S	0 (0)	1 (100)	-	2 (33.33)	3 (100)	-	6 (100)	4 (100)	4 (100)	1 (100)	2 (100)	-
		I	0 (0)	0 (0)	-	0 (0)	0 (0)	-	0 (0)	0 (0)	0 (0)	0 (0)	0 (0)	-
		R	5 (100)	0 (0)	-	4 (66.67)	0 (0)	-	0 (0)	0 (0)	0 (0)	0 (0)	0 (0)	-
*Methicilin-Resistent Staphylococcus Aureus*	5	S	-	0 (0)	1 (100)	1	4 (80)	1 (33.33)	2 (66.67)	1 (100)	3 (100)	0 (0)	1 (33.33)	1 (100)
		I	-	0 (0)	0 (0)	0	0 (0)	0 (0)	0 (0)	0 (0)	0 (0)	0 (0)	1 (33.33)	0 (0)
		R	-	1 (100)	0 (0)	2	1 (20)	2 (66.67)	1 (33.33)	0 (0)	0 (0)	3 (100)	1 (33.33)	0 (0)
*Enterococcus* spp.	4	S	1 (33.33)	-	-	0	3 (100)	-	1 (100)	3 (100)	-	1 (50)	-	3 (75)
		I	0 (0)	-	-	0	0 (0)	-	0 (0)	0 (0)	-	0 (0)	-	0 (0)
		R	2 (66.7)	-	-	0	0 (0)	-	0 (0)	0 (0)	-	1 (50)	-	1 (25)
*Proteus* spp.	6	S	1 (33.33)	0 (0)	-	1	5 (100)	-	5 (100)	5 (100)	1 (100)	0 (0)	2 (100)	-
		I	0 (0)	0 (0)	-	0	0 (0)	-	0 (0)	0 (0)	0 (0)	1 (100)	0 (0)	-
		R	2 (66.67)	1 (100)	-	1	0 (0)	-	0 (0)	0 (0)	0 (0)	0 (0)	0 (0)	-
*Corynebacterium macginleyi*	2	S	-	-	-	-	2 (100)	-	-	-	-	2 (100)	-	1 (100)
		I	-	-	-	-	0 (0)	-	-	-	-	0 (0)	-	0 (0)
		R	-	-	-	-	0 (0)	-	-	-	-	0 (0)	-	0 (0)
*Pseudomonas aeruginosa*	3	S	0 (0)	-	-	-	3 (100)	-	2 (100)	1 (100)	1 (100)	-	2 (100)	-
		I	0 (0)	-	-	-	0 (0)	-	0 (0)	0 (0)	0 (0)	-	0 (0)	-
		R	1 (100)	-	-	-	0 (0)	-	0 (0)	0 (0)	0 (0)	-	0 (0)	-
*Streptococcus pyogenes (Beta-hemolytic)*	3	S	-	-	-	-	1 (100)	2 (100)	1 (100)	1 (100)	-	1 (50)	-	2 (100)
		I	-	-	-	-	0 (0)	0 (0)	0 (0)	0 (0)	-	0 (0)	-	0 (0)
		R	-	-	-	-	0 (0)	0 (0)	0 (0)	0 (0)	-	1 (50)	-	0 (0)
*Escherichia coli*	2	S	0 (0)	1 (50)	1 (100)	1 (100)	2 (100)	-	2 (100)	2 (100)	2 (100)	2 (100)	1 (100)	-
		I	0 (0)	0 (0)	0 (0)	0 (0)	0 (0)	-	0 (0)	0 (0)	0 (0)	0 (0)	0 (0)	-
		R	2 (100)	1 (50)	0 (0)	0 (0)	0 (0)	-	0 (0)	0 (0)	0 (0)	0 (0)	0 (0)	-
*Enterobacter* spp.	1	S	0 (0)	1 (50)	-	-	-	-	1 (100)	-	-	-	-	-
		I	0 (0)	0 (0)	-	-	-	-	0 (0)	-	-	-	-	-
		R	1 (100)	0 (0)	-	-	-	-	0 (0)	-	-	-	-	-
*Haemophilus* spp.	2	S	1 (50)	0 (0)	-	-	2 (100)	-	-	1 (100)	1 (100)	1 (100)	-	-
		I	0 (0)	0 (0)	-	-	0 (0)	-	-	0 (0)	0 (0)	0 (0)	-	-
		R	1 (50)	1 (100)	-	-	0 (0)	-	-	0 (0)	0 (0)	0 (0)	-	-
*Serratia marcescens*	1	S	-	-	-	0 (0)	1 (100)	-	1 (100)	1 (100)	-	-	1 (100)	-
		I	-	-	-	0 (0)	0 (0)	-	0 (0)	0 (0)	-	-	0 (0)	-
		R	-	-	-	1 (100)	0 (0)	-	0 (0)	0 (0)	-	-	0 (0)	-

AS—antibiotic sensitivity, S—bacteria sensitive to antibiotic, I—bacteria with intermediate sensitivity to antibiotic, R—bacteria resistant to antibiotic; AMP—ampicillin, AMX—amoxicillin, CLR—clarithromycin, CXM—cefuroxime, CIP—ciprofloxacin, DOX—doxycycline, ERY—erythromycin, GEN—gentamycin, LVX—levofloxacin, OFX—ofloxacin, TET—tetracycline, TOB—tobramycin, VAN—vancomycin.

**Table 2 diagnostics-13-03409-t002:** Susceptibility of all types of bacteria to antibiotics.

Type of Antibiotic Tested	Abbrev.	Total Number of Bacteria	Response of Bacteria to Antibiotic					
			S	I	R	S %	I %	R %
Vancomycin	VAN	46	44	1	1	97.8	2.2	0
Ceftriaxone	CRO	24	18	2	4	75	8.3	16.7
Cefuroxime	CXM	50	35	1	14	70	2	28
Cefazolin	CFZ	3	1	0	2	33.3	0	66.7
Meropenem	MEM	17	16	0	1	94.1	0	5.9
Imipenem	IPM	15	14	0	1	93.3	0	6.7
Moxifloxacin	MXF	39	32	4	3	82.1	7.7	10.3
Levofloxacin	LVX	68	60	4	4	88.2	5.9	5.9
Ofloxacin	OFX	38	33	1	4	86.8	2.6	10.5
Ciprofloxacin	CIP	117	100	7	10	85.5	6	8.5
Netilmicin	NET	20	19	1	0	95	5	0
Tobramycin	TOB	58	44	8	6	75.9	13.8	10.3
Amikacin	AMK	20	19	0	1	95	0	5
Kanamycin	KAN	9	6	0	3	66.7	0	33.3
Gentamicin	GEN	121	99	3	19	81.8	2.5	15.7
Chloramphenicol	CHL	54	49	1	4	90.7	7.4	1.9
Tetracycline	TET	77	39	4	34	50.6	5.2	44.2
Doxycycline	DOX	24	14	0	10	58.3	0	41.7
Rifampicin	RIF	41	38	1	2	92.7	2.4	4.9
Azithromycin	AZM	19	6	0	13	31.6	0	68.4
Clarithromycin	CLR	36	19	0	17	52.8	0	47.2
Erythromycin	ERY	100	44	6	50	44	6	50
Ampicillin	AMP	45	7	0	38	15.6	0	84.4
Amoxicillin	AMX	20	19	0	1	95	0	5

S—bacteria sensitive to antibiotic, I—bacteria with intermediate sensitivity to antibiotic, R—bacteria resistant to antibiotic; S %—percentage of all-type bacteria sensitive to antibiotic, I %—percentage of all-type bacteria with intermediate sensitivity to antibiotic, R %—percentage of all-type bacteria resistant to antibiotic.

**Table 3 diagnostics-13-03409-t003:** Susceptibility of Gram-positive bacteria to antibiotics.

Type of Antibiotic Tested	Abbrev.	Total Number of Gram-Positive Bacteria	Response of Bacteria to Antibiotic					
			S	I	R	S %	I %	R %
Vancomycin	VAN	45	44	1	0	97.8	2.2	0
Ceftriaxone	CRO	15	9	2	4	60	13.3	26.7
Cefuroxime	CXM	39	30	1	8	76.9	2.6	20.5
Cefazolin	CFZ	1	0	0	1	0	0	100
Meropenem	MEM	11	10	0	1	90.9	0	9.1
Imipenem	IPM	12	11	0	1	91.7	0	8.3
Moxifloxacin	MXF	36	29	3	4	80.6	8.3	11.1
Levofloxacin	LVX	54	46	4	4	85.2	7.4	7.4
Ofloxacin	OFX	28	23	1	4	82.1	3.6	14.3
Ciprofloxacin	CIP	99	82	7	10	82.8	7.1	10.1
Netilmicin	NET	19	18	1	0	94.7	5.3	0
Tobramycin	TOB	48	35	7	6	72.9	14.6	12.5
Amikacin	AMK	10	9	0	1	90	0	10
Kanamycin	KAN	8	5	0	3	62.5	0	37.5
Gentamicin	GEN	102	80	3	19	78.4	2.9	18.6
Chloramphenicol	CHL	47	42	1	4	89.4	2.1	8.5
Tetracycline	TET	70	33	3	34	47.1	4.3	48.6
Doxycycline	DOX	19	9	0	10	47.4	0	52.6
Rifampicin	RIF	38	36	1	1	94.7	2.6	2.6
Azithromycin	AZM	18	5	0	13	27.8	0	72.2
Clarithromycin	CLR	35	18	0	17	51.4	0	48.6
Erythromycin	ERY	99	44	6	49	44.4	6.1	49.5
Ampicillin	AMP	29	5	0	24	17.2	0	82.8
Amoxicillin	AMX	19	9	0	10	47.4	0	52.6

S—Gram-positive bacteria sensitive to antibiotic, I—Gram-positive bacteria with intermediate sensitivity to antibiotic, R Gram-positive bacteria resistant to antibiotic; S %—percentage of Gram-positive bacteria sensitive to antibiotic, I %—percentage of Gram-positive bacteria with intermediate sensitivity to antibiotic, R %—percentage of Gram-positive bacteria resistant to antibiotic.

**Table 4 diagnostics-13-03409-t004:** Susceptibility of Gram-negative bacteria to antibiotics.

Type of Antibiotic Tested	Abbrev.	Total Number of Gram-Negative Bacteria	Response of Bacteria to Antibiotic					
			S	I	R	S %	I %	R %
Vancomycin	VAN	1	0	0	1	0	0	100
Ceftriaxone	CRO	9	9	0	0	100	0	0
Cefuroxime	CXM	11	5	0	6	45.5	0	54.5
Cefazolin	CFZ	2	1	0	1	50	0	50
Meropenem	MEM	6	6	0	0	100	0	0
Imipenem	IPM	3	3	0	0	100	0	0
Moxifloxacin	MXF	3	3	0	0	100	0	0
Levofloxacin	LVX	14	14	0	0	100	0	0
Ofloxacin	OFX	10	10	0	0	100	0	0
Ciprofloxacin	CIP	18	18	0	0	100	0	0
Netilmicin	NET	1	1	0	0	100	0	0
Tobramycin	TOB	10	9	1	0	90	10	0
Amikacin	AMK	10	10	0	0	100	0	0
Kanamycin	KAN	1	1	0	0	100	0	0
Gentamicin	GEN	19	19	0	0	100	0	0
Chloramphenicol	CHL	7	7	0	0	100	0	0
Tetracycline	TET	7	6	1	0	85.7	14.3	0
Doxycycline	DOX	5	5	0	0	100	0	0
Rifampicin	RIF	3	2	0	1	66.7	0	33.3
Azithromycin	AZM	1	1	0	0	100	0	0
Clarithromycin	CLR	1	1	0	0	100	0	0
Erythromycin	ERY	1	0	0	1	0	0	100
Ampicillin	AMP	16	2	0	14	12.5	0	87.5
Amoxicillin	AMX	6	3	0	3	50	0	50

S—Gram-negative bacteria sensitive to antibiotic, I—Gram-negative bacteria with intermediate sensitivity to antibiotic, R—Gram-negative bacteria resistant to antibiotic; S %—percentage of Gram-negative bacteria sensitive to antibiotic, I %—percentage of Gram-negative bacteria with intermediate sensitivity to antibiotic, R %—percentage of Gram-negative bacteria resistant to antibiotic.

## Data Availability

Data available on request due to restrictions eg privacy or ethical. The data presented in this study are available on request from the corresponding author. The data are not publicly available due to the clinic policy on patient database.

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
