# Peer review of "Susceptibility of Ocular Surface Bacteria to Various Antibiotic Agents in a Romanian Ophthalmology Clinic"

_diagnostics, 2023, doi:10.3390/diagnostics13223409_

Round 1

Reviewer 1 Report

Comments and Suggestions for Authors

Title “

 Susceptibility of ocular surface bacteria to various antibiotic 2 agents” add “from a Romanian Hospital”  or similar

Change “

 most of the contamination flora is “ to “

 many strains of the microbiota that cause infections are”

“Staphylococcus aureus, coagulase-50 negative Staphylococcus and Corynebacterium “ - microbial names should be in italics

“flora “ they are not “flowers/plants” a better word in “microbiota”

“in iatrogenic eye infections. The method of” - split to a new paragraph here “The method of …”

Materials and methods

Where was the study conducted? Which hospital or university? Which country (I assume Romania)

Results

“tested probes “ what are “probes”?

Bacterial names should be in italics

Figure 1 does not appear to include any data

“an increased resistance “ suggests that this increased either over time or in comparison with some other antibiotic - I think you might mean “a high level of resistance” - and if so -0 then also change abstract “

 we observed an increased resistance of the cefuroxime for both Gram-positive and negative bacteria” to “

 we 22 observed a high level of resistance of the cefuroxime for both Gram-positive and negative bacteria “

Comments on the Quality of English Language

not applicable

Reviewer 2 Report

Comments and Suggestions for Authors

This article shows the resistance patterns of ocular bacteria in Romania. The relevance of this article is related to the global microbial resistance to antibiotics and their association with clinical practices.   Despite their obvious importance to global health, the authors  must  address some issues:

Major observations:

The authors used the word “flora” in some parts of the manuscript. Currently, the appropriate term is microbiota; please correct it.

Methods. How many laboratories contributed to the data? Are all in the same clinical center? Have they used the same identification technique or laboratory equipment to perform the sensibility/resistance assays? What do you mean by an “approved laboratory”?

Intermediate inhibition was considered according to the maximum doses for each antibiotic? If yes, clarify this issue; not all readers are experts in microbiology.

Discussion. How do you associate the ophthalmological practices of antimicrobial use as a determinant of your results? Or is it possible to explain your results observed with cefuroxime and its impact on the population?,

Can you use your data comparing results with other reference centers from other countries to study or help pinpoint regional differences and similarities, contributing to a more comprehensive global strategy in the fight against microbial resistance? Or in the standardization of clinical ophthalmological practices?

Minor observations:

Figure 1. I can't see in the figure the distribution of the bacterial isolates; please check and correct it.

Tables. Information in some tables could be clearer to read; please check the letter size or distribution.
